# The Nursing Stress Scale-Spanish Version: An Update to Its Psychometric Properties and Validation of a Short-form Version in Acute Care Hospital Settings

**DOI:** 10.3390/ijerph17228456

**Published:** 2020-11-15

**Authors:** Ana María Porcel-Gálvez, Sergio Barrientos-Trigo, Sara Bermúdez-García, Elena Fernández-García, Mercedes Bueno-Ferrán, Bárbara Badanta

**Affiliations:** 1Faculty of Nursing, Physiotherapy and Podiatry, University of Seville, 41009 Seville, Spain; aporcel@us.es (A.M.P.-G.); efernandez23@us.es (E.F.-G.); mbueno2@us.es (M.B.-F.); bbadanta@us.es (B.B.); 2Research Group under the Andalusian Research, Development and Innovation Scheme CTS-1019 Complex Care, Chronic and Health Outcomes, Instituto de Biomedicina de Sevilla (IBIS), 41013 Seville, Spain; 3Hospital Vithas Sevilla, Avda. Plácido Fernández Viagas, s/n PC, Castilleja de la Cuesta, 41950 Seville, Spain; saraber17@gmail.com; 4Hospital Virgen del Rocío, Av. Manuel Siurot, S/n, 41013 Seville, Spain

**Keywords:** hospital, nursing staff, occupational stress, working conditions, assessment tools, validation, instrument development, psychometric assessment, questionnaires

## Abstract

Stressful working conditions are correlated with a negative impact on the well-being of nurses, job satisfaction, quality of patient care and the health of the staff. The Nursing Stress Scale (NSS) has been shown to be a valid and reliable instrument to assess occupational stressors among nurses. This study updates the psychometric properties of the “NSS-Spanish version” and validates a short-form version. A cross-sectional design was carried out for this study. A reliability analysis and a confirmatory factor analysis and an exploratory factor analysis were undertaken. Items were systematically identified for reduction using statistical and theoretical analysis. Correlation testing and criterion validity confirmed scale equivalence. A total of 2195 Registered Nurses and 1914 Licensed Practical Nurses were enrolled. The original 34-item scale obtained a good internal consistency but an unsatisfactory confirmatory and exploratory factor analysis. The short-form Nursing Stress Scale (11-items) obtained a good internal consistency for Registered Nurses (α = 0.83) and for Licensed Practical Nurses (α = 0.79). Both Nursing Stress Scales obtained a strong correlation for Registered Nurses (rho = 0.904) and for Licensed Practical Nurses (rho = 0.888). The 11-item version of the Nursing Stress Scale is a valid and reliable scale to assess stress perception among Registered Nurses and Licensed Practical Nurses. Its short-form nature improves the psychometric properties and the feasibility of the tool.

## 1. Introduction

The deterioration of the well-being of nurses linked to complex patient care has led non-English speaking countries to validate the Nursing Stress Scale (NSS) in other languages to measure potentially stressful situations and compare their results with those of other countries. Therefore, this instrument has been validated in Chinese [1], Bahasa Melayu [2], Indian [3], and Spanish [4]. The NSS Spanish version was developed from the original tool consisting of 34 items (34 potentially stressful situations). After the cultural adjustment, it was validated on a random sample of 201 nurses (103 nurses and 98 nursing assistants) in fourteen units of a public hospital in Spain [5]. In all cases, the units were chosen because the patients presented a range of medical conditions requiring different types of nursing care and exposing nurses to various sources of stress.

The Nursing Stress Scale (NSS) was introduced by Gray and Anderson for the first time in 1981 through a literature review and interviews with nurses, doctors and chaplains. It was administered to 122 nurses from five clinical units of a private general hospital, in order to measure the occurrence and frequency of certain situations perceived as stressful by the nursing staff [6]. 

### Background

Stress, as an internal cue in the physical, psychological or social environment that threatens the equilibrium of an individual, has a great impact in terms of work performance [7]. Work-related factors affect the employees, changing their psychological and physiological condition so that the person is unable to function as normal [8]. The context in which nursing takes place makes it a high-risk profession for burnout [9]. Exposure to pain and death or lack of support from clinical managers can affect nurses physically and psychologically [9,10,11,12]. Other stressful work factors are accelerated work, constant handling of emergency situations, uncertainty about treatments, confrontation with family members, conflict between the responsibilities assumed and the little authority granted [13,14]. To this is added the decrease in the number of nurses and the demands caused by the chronicity and ageing of the population [15]. All this generates stress for nurses, affects their psychological well-being and job satisfaction, enhances absenteeism and abandonment of the profession [16,17] and finally, negatively affects the quality of patient care and the health of the staff. 

There are instruments that measure the same concept of “stress”, such as the Depression Anxiety Stress Scales (DASS) [18], General Health Questionnaire (GHQ) [19], Perceived Stress Scale (PSS) [20], Karasek Job Content Questionnaire (JCQ) [21], Occupational Stress in Remote Area Nursing: Development of the Remote Area Nursing Stress Scale (RANSS) [22], or Traumatic and Routine Stressors Scale on Emergency Nurses (TRSS-EN) [23]. However, they have certain limitations: some of them are not designed for the work context (e.g., DASS, GHQ) and others, although they assess the work environment, are not specifically adapted to the healthcare environment, such as the PSS and JCQ questionnaires. The TRSS-EN only evaluates the stressor elements in the nursing staff in a specific setting, the emergency department, as with the RANSS in the setting of health centers. On the other hand, the DASS questionnaire and GHQ are made up of situations that not only evaluate stress, but also depression, anxiety and other disorders of minor psychiatric origin, without a real division between the factors that they evaluate. 

Despite the existence of different general scales to measure stress, an instrument specifically designed for nurses is necessary to design, implement and evaluate interventions aimed at preventing and controlling occupational stressors among nurses as a high-risk population. In this context, the “Nurse Stress Checklist” [24], “Nurse Stress Index” [25] and “Nursing Stress Scale” (NSS) [1] stand out. All of them have a reliability greater than 0.85 and they can be used in general settings (surgical and medical units), allowing a greater comparison between nursing professionals. Although the largest sample size was used to validate the NSI, it presented other weaknesses compared to the NSS. The variance explained in the Exploratory Factor Analysis is inadequate (45.6%) and lower than that of the NSS. The NSS stands out with respect to the previous scales mentioned, showing an explained variance of 85%. Furthermore, it has a similar number of items (34), and adequate reliability both on the global scale (coefficient α of 0.89) and between factors (0.64–0.80). Finally, the NSS is the scale with the largest number of factors (seven subscales), some of which address similar categories to other stress scales (e.g., Lack of support, Conflict with Physicians/other nurses, and Inadequate preparation), and it also provides new aspects to be considered, such as “Death and dying”.

Moreover, the NSS has been one of the most used scales in recent years to assess occupational stressors among nurses. Since 2015 it has been used to measure work stress in areas such as oncology [26], neonatal intensive care units [27], pediatrics [28], psychiatrics [29] and other general services [30]. The relationship with other important criteria to which stress is theoretically related, namely, trait anxiety, state anxiety, job satisfaction, and turnover was also studied. Nursing stress scores were found to be positively correlated with trait anxiety among hospital nursing staff and inversely correlated with job satisfaction [1].

The last adaptation to the Spanish language was recorded 21 years ago, and since then no study has updated the psychometric properties. The assessment instruments need to be updated periodically, since the construct for which they were developed may change over time. Furthermore, validating instruments that are shorter is useful to improve the response rates and the feasibility. In relation to stress, some authors claim the need to intervene on stressors since the stress load can increase, decrease or change depending on the temporal context in which they develop [31]. As a result, it is necessary to update the instruments for the detection of stressors to meet the requirements of a more prepared, efficient and autonomous nursing team [32].

Therefore, the purpose of the study was to update the psychometric properties of the “Nursing Stress Scale-Spanish version” and validate a short-form version in nursing staff in acute care hospital settings in the Public Health Service in Spain.

## 2. Materials and Methods 

### 2.1. Study Design

A multicenter, cross-sectional, and psychometric validation study was developed. The Consensus-based Standards for the selection of health Measurement Instruments (COSMIN) checklist was used to guide the study (See Appendix A). The COSMIN checklist is recommended for designing studies to evaluate the measurement properties of existing patient-reported outcome measures [33]. 

### 2.2. Setting and Sample

The sample consisted of registered nurses (RN) and licensed practical nurses (LPN) working at nineteen public hospitals in Andalusia belonging to the Spanish National Health System (which provides universal and free coverage to 8.4 million inhabitants in the region). These hospitals were classified according to their level of specialization and reference population in primary (>500 beds and large metropolitan areas), specialist (between 200 and 500 beds and small metropolitan areas), and tertiary hospitals (<200 beds and rural areas). Nursing staff constituted 45.5% of the hospital workforce in the Public Health Services. Currently, in Spain there are RNs who studied a diploma in nursing (3 academic years) and, since 2010, the RNs have had a degree in nursing (4 academic courses). In addition, their competence level is higher and more independent than the LPN. LPN is a two-year vocational training and they carry on basic nursing tasks.

Sampling was carried out in two phases. In the first phase, convenience sampling was used, selecting the participating units from the 19 hospitals to guarantee the homogeneity and representativeness of the main settings in hospitals. These units were Internal Medicine, General Surgery, Traumatology, Cardiology, Neurology, Pneumology, Urology, Otorhinolaryngology, Infectious Diseases, Oncology and Palliative Care. In the second phase, the sample was drawn from the Official Data Sources of Hospitals of 17,717 RN and 13,513 LPN. All the nursing staff were included, except those who were not active practitioner nurses during the study period (due to being on holiday or because of a work disability). 

### 2.3. Data Collection

Data collection took place between October 2017 and April 2019. The recruitment of the Registered Nurses and Licensed Practical Nurses was carried out in two phases. In the first phase, the head researcher contacted the nursing directors of each of the participating hospitals. In the second phase, RNs and LPNs were recruited from participating wards by the nursing managers. The questionnaires were mailed to the participating hospitals and delivered to nurses. The questionnaires were delivered by the nursing managers to the RNs and LPNs and were self-administered. The coding of the questionnaires prevented the professional participated twice in the study. The participants were asked about their demographic characteristics and the Nursing Stress Scale-Spanish version. 

Once nurses received the questionnaires, they had two weeks to answer. The questionnaires were then put in an envelope and handed to the nursing manager. When all the questionnaires had been collected, the nursing manager sent them by post to the research team. All returned questionnaires were previewed and registered in the platform by the researchers, guaranteeing at all times the anonymity of the professionals who participated. 

In the participating units, the staff amounted to 7639 Registered Nurses and Licensed Practical Nurses. In total, 4109 subjects completed the survey. The response rate was 52.5% (Figure 1). The non-respondents (47.5%) were 58.7% RN and 41.3% LPN. According to hospital admission, primary hospitals (41.6%), specialities (25.2%) and tertiary (33.3%). Most worked in medical units (57.4%) than surgical (42.6%). 

### 2.4. Measures

Descriptive variables: gender (male and female), age, level of education (vocational training, registered nurse and bachelor in nursing sciences, master in nursing sciences, PhD in nursing science), years of professional experience in the current job and in the current service, hospital admission (primary, specialist and tertiary), health unit/service (surgical and medical), contract (permanent and temporary), workday (full-time (35–40 h/week) and part-time), and shift hours.

Instrument: the present survey used the Nursing Stress Scale-Spanish version; a self-administered anonymous questionnaire. Validation was conducted on a random sample of 201 health professionals (103 RN and 98 LPN) by Más and Escribà (1998), and the construct validity was obtained through the correlation of the scale with the 7 dimensions of the Health questionnaire SF-36 and 28-item version of Goldberg’s General Health Questionnaire. The Cronbach’s alpha coefficient is 0.92 for the total scale and the internal consistency coefficients of the sub-scales range between 0.83 and 0.49, and the total explained variance is 63.4% [5].

Each of the 34 items of the NSS have been grouped into seven areas of job-related stress (F1 = 7 items; FII = 5 items; FIII = 3 items; FIV = 3 items; FV = 5 items; FVI = 6 items; FVII = 5 items), corresponding to the physical, psychological and social spheres: (1) Physical Environment (workload); (2) Psychological Environment (death and suffering, inadequate training, lack of support, and uncertainty about treatments); and (3) Social Environment in the hospital (conflict with physicians and conflict with other nurses). The authors found only one irrelevant item as part of the workload: “Breakdown of computer”, and it was replaced by “Frequent job interruptions” [5]. 

The answers are based on a 4-point Likert scale (never = 0, sometimes = 1, frequently = 2, and very frequently = 3). The sum of the scores obtained resulted in a global index, ranging from 0 to 102. A higher score reflects a higher level of stressors.

### 2.5. Data Analyses

The SPSS statistical package (version 26.0) was used for statistical analysis (SPSS/IBM, Chicago, IL, USA). Frequencies, percentages, means and standard deviations were calculated for univariate analysis. A normality test was carried out for continuous variables using the Kolmogorov–Smirnov test and, based on the result, the tests were selected as independent t-tests or Mann–Whitney tests. Inferential analysis was performed using Chi-squared tests or Mann–Whitney tests, depending on the nature of the variable (categorical, ordinal or continuous). 

The psychometric analysis included reliability and validity tests. Internal consistency was calculated using Cronbach’s alpha coefficient. To test the factor structure obtained in previous studies, a Confirmatory Factor Analysis (CFA) was used. For this, a model fit evaluation was performed using the root mean square approximation residue (RMSEA), describing a good adjustment of the model with values lower to 0.06 are obtained [34]; the standardized root mean square residue (SRMR) with values lower to 0.08 [35]. The *p*-value of the closeness of fit (PCLOSE) and the Chi-square ratio and the number of degrees of freedom (χ^2^/DF) was calculated considering it acceptable if the value resulting was higher than 4 [36]. The Comparative Fit Index (CFI), Parsimony Index (NFI), Goodness of Fit Index (GFI), Incremental Fit Index (IFI), Parsimony Normalised Fit Index (PNFI) and Parsimony Comparative Fit Index (PCFI) were also calculated, and their values had to be higher than 0.90 [37].

Additionally, an exploratory factor analysis (EFA) was undertaken, with Principal Component Analysis and varimax rotation. Kaiser–Meyer–Olkin (KMO) and Bartlett’s sphericity test had already been performed to determine if the data were suitable for factor analysis. A global analysis was carried out for each subsample (RN and LPN). A short-form version was validated based on the highest corrected item-total correlation within each subscale using a correlation coefficient ≥0.4 [38] and removing items with low factor loadings (<0.5). A new 11-item version was achieved. Regarding the convergent validity, NSS-34 was used as the gold standard for NSS-11 [33]. The Spearman’s Rho correlations coefficient was used, following the hypothesis of a strong and positive relationship. 

### 2.6. Ethical Considerations

The present study was approved by the Regional Ethics Committee (CPMP/ICH/135/95). All participants received an information sheet that illustrated the study and completed an informed consent form. In addition, anonymity and respect for privacy were ensured. 

## 3. Results

### 3.1. Sample Characteristics

To analyze the psychometric properties of the NSS, 2195 RN and 1914 LPN were enrolled. From the final 4109 participants, 85% were women. Their average age was 46.5 years old, 91.2% of RNs held a 3 or 4-year Bachelor’s Degree in Nursing, while the remaining 8.8% had studied an additional degree in other disciplines. Among the LPNs, 54.5% had only accessed the accredited practical nursing program and 6.2% had university studies. The average years of experience was 18 years and seniority in the unit was 7.6 years. Most worked full time (88%), in shifts with an average of 9.1 h. Moreover, there were no statistical differences between hospital admission, health service and shift hours (*p* > 0.05). However, there were significant differences according to gender, age, level of education, years of professional experience, and contract (*p* < 0.05). More details on the characteristics of the participants are shown in Table 1.

### 3.2. Descriptive Analysis of NSS-34 

The item score ranged from 0.32 (SD = 0.61) for “Problems with a supervisor” to 2.41 (SD = 1) for “Frequent job interruptions” (Table 2).

### 3.3. Validation of NSS-34

The original 34-item scale obtained a Cronbach’s alpha of 0.917 (95% Confidence Interval (CI) [0.912;0.922] for RN and 0.904 (95% CI [0.897;0.91]) for LPN. The CFA was performed on this seven-factor model (Figure 2). The seven factors were: (1) Death and suffering (7 items), (2) Conflict with physicians (5 items), (3) Inadequate training (3 items), (4) Lack of support (3 items), (5) Conflict with other nurses (5 items), (6) Workload (6 items), (7) Uncertainty about treatments (5 items). The fitness of this model yielded the following values: RMSEA= 0.058; SRMR = 0.052; PCLOSE < 0.01; CMIN/DF = 15.018 (*p* < 0.01); CFI = 0.843; NFI = 0.834; IFI = 0.843; PNFI = 0.752; PCFI = 0.761.

The EFA showed a 7-factor solution with 54.7% of the total being explained variance for RN and 50.3% for LPN. The result of the KMO was 0.932, with factor loads ranging from 0.326 to 0.763 for RN, and 0.92 with factor loads ranging from 0.345 to 0.736 for LPN. In addition, Bartlett’s Test of sphericity was significant in both cases (*p* < 0.001), thus confirming the adequacy of the EFA.

### 3.4. Validation of Short-form NSS-11

A short-form version was validated, based on the highest corrected item-total correlation within each subscale (≥0.4) and removing the items with low factor loadings (<0.5). The new version consisting of 11 items (NSS-11) improved the psychometric properties of the original scale. The data showed a Cronbach’s alpha of 0.83 (95%CI[0.818;0.839]) for RN and 0.79 (95%CI[0.776;0.804]) for LPN. The results of KMO and Bartlett’s Test of sphericity were appropriate for AFE, both for RN (0.803; *p* < 0.001) and LPN (0.781; *p* < 0.001). In both groups, the AFE maintained the same 4-factor structure with 69.7% of the total variance explained in RN, and 65.6% of the total variance explained in LPN (Table 3).

Regarding the grouping of the items of each factor, it varied in the different groups. For the scale with RN, the factors were: (1) Workload (α = 0.796); (2) Conflict with physicians and other nurses (α = 0.719); (3) Inadequate preparation to deal with the emotional needs of patients and their families (α = 0.773); (4) Lack of support (α = 0.754). In the case of the LPN scale, they were: (1) Workload and lack of support (α = 0.767); (2) Conflict with nurses or supervisors (α = 0.77); (3) Inadequate preparation to deal with the emotional needs of patients and their families (α = 0.612); (4) Conflict with physicians (α = 0.604).

### 3.5. Comparison of the 34-Item NSS and NSS-11 (Short-Form)

The two Nursing Stress Scale versions (the 34-item and the new short-form versions) obtained a high correlation between them: r = 0.904 (*p* < 0.001) for RN and r = 0.888 (*p* < 0.001) for LPN. The mean value of NSS-34 was 39 (SD = 14.3), and for NSS-11 it was 12.4 (SD = 5.3) (Figure 3).

### 3.6. COSMIN Checklist

The quality of the procces was assessed by COSMIN checklist, obtaining the following scores: General: Very good; Content validity: NR; Structural validity: Adequate; Internal consistency: Very good; Cross-cultural validity/measurement invariance: NR; Measurement error and reliability: Very good; Criterion validity: Very good; Hypotheses testing for construct validity: (A) Very good, (B) Very good; Responsiveness: Very good; Translation process: NR (Table 4).

## 4. Discussion

The purpose of this study was to update the psychometric properties of the “Nursing Stress Scale (NSS)-Spanish version” and to validate a short-form version in a working population of RN and LPN in the Public Health System in Spain. This has allowed the development of a more parsimonious and feasible tool, due to the reduction of its factors and items (NSS-11) with respect to the original scale of 34 items [6], and a larger sample of RN and LPN compared to the validation of the NSS Spanish version has been used [5]. The results support a general improvement of the psychometric properties of the NSS-11 compared to the original and Spanish version, which guarantees having a reliable and up-to-date tool for the present time. In addition, a brand new aspect that has been added was evaluating the quality of the validation of the scale through the COSMIN checklist [32].

The feminisation that characterises the nursing profession is also reflected in our study [39]. Results from other studies show that female nurses’ paid work schedules make meeting their responsibilities virtually impossible. Thus, work stress coupled with their gender role generates pressure due to familial and partner dependency, all of them cultural and social norms which assign women primary responsibility for unpaid work [40]. On the other hand, authors emphasise the need for interventions to reduce stress and burnout taking the ages of nursing staff into account [41]. A study with nurses in one tertiary hospital in Vietnam showed that the lowest proportion of self-perceived stress, anxiety and depression were observed in the youngest nurses, who had fewer years of service [42].

In this study, the factors perceived as less stressful for professionals are problems with supervisors and doctors, difficulty working with colleagues and receiving criticism from them. However, factor 1 “Workload”, items 28, 30 and 34 had the highest scores for RN and LPN, which is also supported by other authors: frequent job interruptions, lack of personnel, patient suffering and not having time to give them emotional support [43]. Similar results were reported by other authors, where workload, interpersonal conflicts, dealing with patients and relatives, with death and dying, and suffering, constitute the factors of the Work Stressor Inventory for Nurses in Oncology [41]. Tran et al. (2019) showed that the stressors to which nurses are exposed in their daily work increase the risk of developing mental disorders such as stress, anxiety and depression [42]. In their study, they detected 18.5% of stress in nurses, and related it to a high demand for tasks and conflicts at work, and a low work reward.

From the results, the updated validation of the NSS-Spanish version (34-items) is closer to the original version, since it goes back to being a tool with 7 factors, ultimately consisting of four factors in the reduced version (NSS-11). Compared to the original NSS, the “Uncertainty Concerning Treatment” and “Death and Dying” factors were removed, while others have been reunited proving better factor loads in our study. In the scale for RN, conflict with other colleagues included both nurses and doctors, while in the case of LPN, two factors appear to differentiate between problems with nurses and supervisors, and conflicts with doctors (probably related to professional competence). In Spain, nurses are the link between doctors and LPN, while LPN maintain closer contact with nurses than with doctors. [44,45]. Despite a substantial reduction in items and factors with loading scores under 0.5 in the final factorial structure, reliability continued to be high and the percentage of total variance being explained increased, surpassing even the first validation of the Spanish scale (63.4%) [5].

Currently, no other study has performed a validation of the NSS in the Spanish environment, although it has occurred in other contexts. Alkrisat and Alatrash (2017) examined the Extended Nursing Stress Scale (ENSS) in acute care settings in California [46]. Kim et al. (2015) evaluated the validity and reliability of the Korean version of the ENSS (ENSS-KV) [47], and Pathak, Chakraborty, and Mukhopadhyay (2013) modified the structure of the NSS and a new scale was named as Modified Nursing Stress Scale (MNSS) for the Indian nursing population [3]. The alpha coefficient of the NSS-11 in this study was found to be 0.815, lower than MNSS, ENSS-KV and ENSS (>0.90). However, unlike the 7 factors of the MNSS and 9 of the ENSS and ENSS-KV, in our study items were shifted according to their load to provide a more meaningful structure, which has allowed the NSS-11 to have 4 factors without conceptual modifications. On the other hand, compared to 34 (MNSS) [3], 48 (ENSS-KV) [47] and 57 (ENSS) [3] items on these scales, the NSS-11 has adequate reliability values, favors the reduction of time to be administered, and can be used for both types of professionals, RN and LPN.

In all cases, the sample was made up only of nurses and the sample size was smaller than that of our study, which places the NSS-11 in an advantageous situation compared to other previous scales. When evaluating supportive interventions directed towards RN and LPN, valid and reliable instruments are needed in order to detect possible changes. While some studies forget LPN in relation to suffering stress [48,49], others show how some factors can contribute to stress and an impaired psychological health status, such as deteriorated management relations and resident, job insecurity and the proximity of residents’ death, correlated with nurses and nurse assistants’ intention to leave [50]. Future studies could explore the relation between stress and other staff outcomes such as job satisfaction, burn-out syndrome or work environment.

Results from the use of this tool focus on the implementation of interventions (micro, meso and macro management level interventions) aimed at improving working conditions to reduce the stress of nursing staff. Some proposals facilitate the reconciliation between work and private life, minimize the interruption of nurses’ tasks, and guarantee the adequate number of personnel taking into account the recommendations of the professional-patient ratio, as well as the psychosocial approach of the patient [51].

### Limitations and Strengths

The main limitation was the response rate (52.5%). Nevertheless, some concerns can be raised about the profile of non-respondent nurses, who could potentially be too stressed to take on the task of filling out the scale. The reduction of the scale could contribute by saving time when completing the survey. This improvement in time could increase the participation of nurses in future studies. In addition, to overcome this limitation, and considering the voluntary nature of participation, the researchers encouraged all staff and provided extensive information on the future implications of this study. This study verified the feasibility of using this instrument as a new stress measurement scale to design and implement strategies and interventions to reduce stressors or mitigate their harmful effects.

## 5. Conclusions

The short version of the NSS-Spanish version has shown a satisfactory level of validity and reliability to assess the perception of stress among nursing staff in Spain. This scale has been reduced to 11 items and 4 factors, making it an easily applicable version in the hospital setting. Reducing the time to complete NSS-11 transforms it into a feasible instrument to visualise stress-inducing situations experienced by nurses. Future research is needed to use the NSS-11 and to conduct a more in-depth assessment of interventions to reduce stress among hospital nurses.

## Figures and Tables

**Figure 1 ijerph-17-08456-f001:**
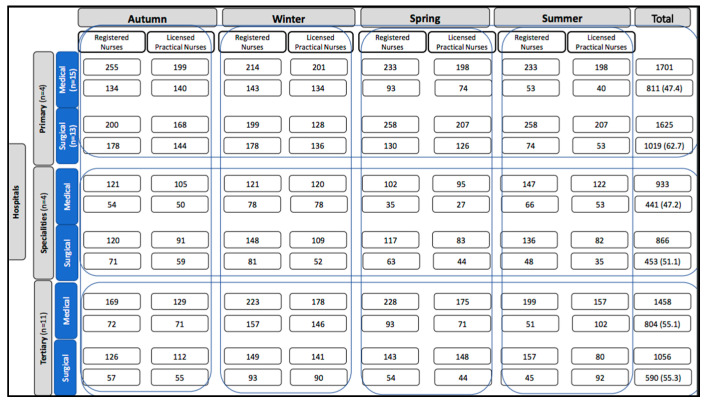
Sampling and recruitment flow chart.

**Figure 2 ijerph-17-08456-f002:**
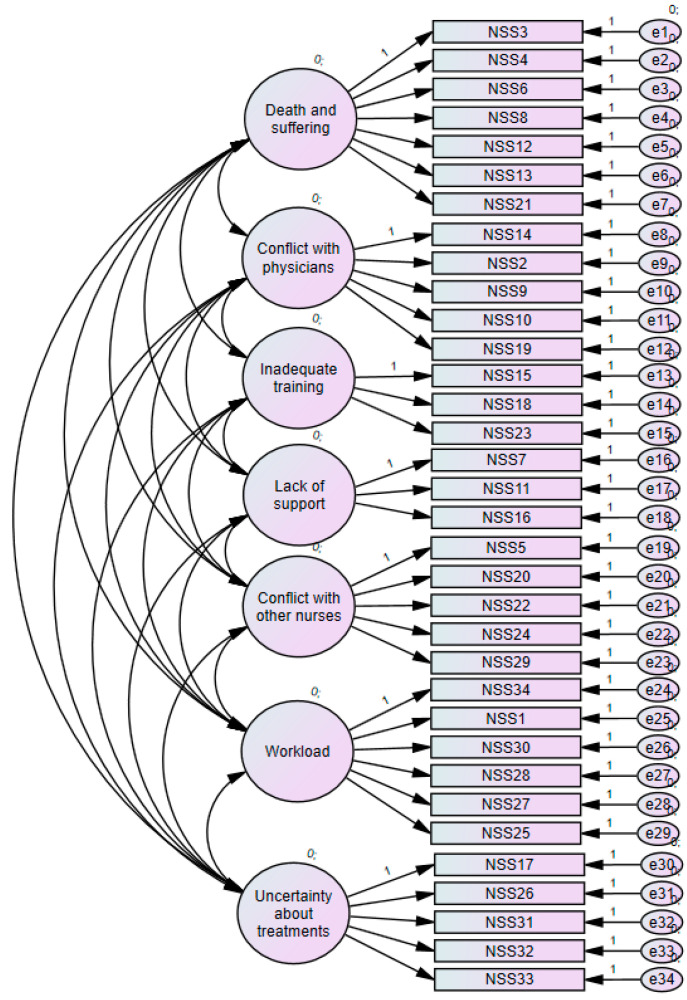
CFA diagram.

**Figure 3 ijerph-17-08456-f003:**
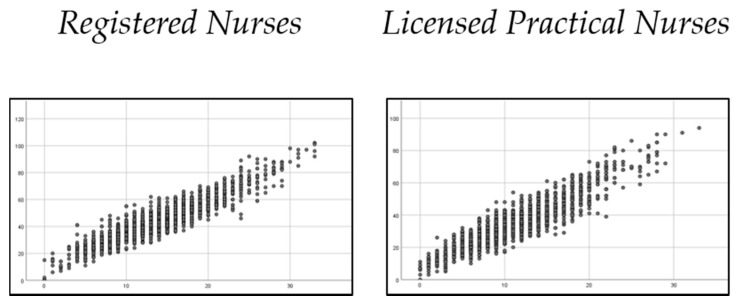
Scatterplot of sum score correlation between 34-item NSS (y-axis) and short-form NSS (x-axis) for registered nurses (RN) and licensed practical nurses (LPN).

**Table 1 ijerph-17-08456-t001:** Sample characteristics.

Variables	RN n (%)	LPN n (%)	Total n (%)	Statistics *p*-Value
	2195 (53.4)	1914 (46.6)	4109	
Gender				<0.001
Male	449 (20)	158 (8)	607 (15)	
Female	1746 (80)	1756 (92)	3502 (85)	
Age (years)				
Mean ± SD	44.3 (8.7)	49 (8.5)	46.5 (8.9)	<0.001
Level of education				<0.001
Vocational training	-	1795 (93.8)	1795 (43.7)	
Registered nurse and Bachelor in nursing sciences	2003 (91.2)	113 (5.9)	2116 (51.5)	
Master in nursing sciences	188 (8.6)	6 (0.3)	194 (4.7)	
PhD in nursing sciences	6 (0.3)	0	6 (0.1)	
Years of professional experience				<0.001
Current job (Mean± SD)	19.4 (8.9)	16.7 (9.3)	18 (9.2)	
Current service (Mean± SD)	8.3 (8.1)	6.7 (6.7)	7.6 (7.5)	
Hospital admission				
Primary	977 (44.5)	842 (44)	1819 (44.2)	
Specialist	494 (22.5)	402 (21)	896 (21.8)	0.230
Tertiary	724 (33)	670 (35)	1394 (34)	
Health Unit/Service				
Surgical	1128 (51.4)	928 (48.5)	2056 (50)	0.066
Medical	1067 (48.6)	986 (51.5)	2053 (50)	
Contract				<0.001
Permanent	1010 (46)	657 (34.3)	1667 (40)	
Temporary	1185 (54)	1257 (65.7)	2442 (60)	
Workday				
Full-time (35–40 h/week)	1821 (83)	1780 (93)	3601 (88)	<0.001
Part-time	374 (17)	134 (7)	508 (12)	
Shift hours				
Mean ± SD	9 (2.7)	9.2 (2.7)	9.1 (2.7)	0.349

**Table 2 ijerph-17-08456-t002:** Description of 34-item Nursing Stress Scale (NSS).

34-ITEM NSS	MEAN ± SD	
1. Frequent job interruptions.	2.41	0.785
2. Criticism by physician.	0.73	0.724
3. Performing procedures that patients experience as painful.	1.45	0.775
4. Feeling helpless when patients fail to improve.	1.23	0.915
5. Conflict with supervisor.	0.32	0.610
6. Listening or talking to a patient about his/her approaching death.	1.02	0.794
7. Lack of opportunity to talk openly with other unit personnel about problems on the unit.	1.13	0.886
8. The death of a patient.	1.47	0.850
9. Conflict with physician.	0.50	0.694
10. Fear of making a mistake in treating a patient.	1.14	0.805
11. Lack of an opportunity to share experiences and feelings with other personnel in the ward/unit.	1.15	0.851
12. The death of a patient with whom you develop a close relationship.	1.30	0.808
13. Physician not being present when patient dies.	1.48	0.988
14. Disagreement concerning the treatment of a patient.	0.92	0.744
15. Feeling inadequately prepared to help with the emotional needs of a patient’s family.	1.02	0.732
16. Lack of opportunity to express to other personnel in the ward/unit my negative feelings towards patient.	0.91	0.766
17. Inadequate information from a physician regarding the medical condition of a patient.	1.31	0.884
18. Being asked a question by a patient for which I do not have a satisfactory answer.	1.11	0.641
19. Making a decision concerning a patient when the physician is unavailable	0.94	0.881
20. Covering other units that are short-staffed.	1.00	1.005
21. Watching a patient suffer.	1.76	0.856
22. Difficulty in working with a particular nurse (or nurse) on the unit.	0.61	0.742
23. Feeling inadequately prepared to help with the emotional needs of a patient.	0.93	0.705
24. Criticism by a superior.	0.56	0.718
25. Unpredictable staffing and scheduling.	1.01	0.874
26. A physician ordering what appears to be inappropriate treatment for a patient.	0.87	0.746
27. Too many non-nursing tasks required (such as administrative tasks).	1.58	0.972
28. Not enough time to provide emotional support to a patient.	1.72	0.876
29. Difficulty in working with a particular nurse (or nurses) within the ward.	0.64	0.747
30. Not enough time to complete all of my nursing tasks.	1.59	0.874
31. A physician not being present in a medical emergency.	1.22	0.893
32. Not knowing what a patient or a patient’s family ought to be told about the patient’s condition and treatment.	1.11	0.774
33. Uncertainty regarding the operation and functioning of specialised equipment.	1.04	0.725
34. Not enough staff to adequately cover the unit.	1.77	0.936

**Table 3 ijerph-17-08456-t003:** Rotated component matrix of 11-item NSS for Registered Nurses and Licensed Practical Nurses.

	Registered Nurses				Licensed Practical Nurses			
	1	2	3	4	1	2	3	4
7. Lack of an opportunity to talk openly with other unit personnel about problems on the unit.				0.787	0.614			
11. Lack of an opportunity to share experiences and feelings with other personnel in the ward/unit.				0.771	0.666			
14. Disagreement concerning the treatment of a patient.		0.570						0.731
15.Feeling inadequately prepared to help with the emotional needs of a patient’s family.			0.889				0.869	
22. Difficulty in working with a particular nurse (or nurses) on the unit.		0.763				0.812		
23.Feeling inadequately prepared to help with the emotional needs of a patient.			0.837				0.769	
26. A physician ordering what appears to be inappropriate treatment for a patient.		0.672						0.712
28. Not enough time to provide emotional support to a patient.	0.769				0.738			
29. Difficulty in working with a particular nurse (or nurses) within the ward.		0.756				0.841		
30. Not enough time to complete all of my nursing tasks.	0.802				0.730			
34. Not enough staff to adequately cover the unit.	0.745				0.729			

Extraction method: Principal component analysis. Rotation method: Varimax with Kaiser normalization.

**Table 4 ijerph-17-08456-t004:** Reporting Guideline Checklist. COSMIN checklist.

Items	Nursing Stress Scale–Spanish Version
General recommendations for the design of a study on measurement properties	+++
Content validity	NR
Structural validity	++
Internal consistency	+++
Cross-cultural validity/measurement invariance	NR
Measurement error and reliability	+++
Criterion validity	+++
Hypotheses testing for construct validity A. Comparison with other outcome measurement instruments (convergent validity) B. Comparison between subgroups (discriminative or known-groups validity)	+++ +++
Responsiveness A. Criterion approach (i.e., comparison to a ‘gold standard’) B. Construct approach (i.e., hypotheses testing; comparison with other outcome measurement instruments) C. Construct approach: (i.e., hypotheses testing: comparison between subgroups) D. Construct approach: (i.e., hypotheses testing: before and after intervention)	+++ +++ +++ NR
Translation process	NR

+++: Very good; ++: Adequate; NR: Not reported.

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
