# Peer review of "The Nursing Stress Scale-Spanish Version: An Update to Its Psychometric Properties and Validation of a Short-form Version in Acute Care Hospital Settings"

_ijerph, 2020, doi:10.3390/ijerph17228456_

Round 1
Reviewer 1 Report
The manuscript presents the results of a study aimed at updating the psychometric properties of the “Nursing Stress Scale - Spanish version” and validating a short-form version in nursing staff in acute care hospital settings in the Public Health Service in Spain. The manuscript is well structured.
The topic is relevant since the valid tools can be extremely useful in improving quality of worker especially for those who work directly with patients. In this sense, it is essential to have tools that allow us to quickly evaluate those constructs that we want to measure.
However, in my opinion, the study presents some methodological problems that need to be addressed and require reorientation on the part of the researchers. I detail these problems below.
Materials and Methods
You have indicated how recruited the participants for approximately two years, guaranteeing the anonymity of the participants. It does not indicate how it has guaranteed that in that period of time the same nurse or graduate has not participated in the study twice
Line 161 describes that the NSS instrument is composed of seven areas, but does not indicate the number of items in each area.
I think it is good that they have carried out an exploratory factor analysis of the 34-item questionnaire but it would be more accurate to include a confirmatory factor analysis. The structure of this questionnaire is well known and it is important to see how it behaves in the study population in order to answer the first objective.
Results
Specify in table 1, if there are missing data (the sum of some of the variables does not give the total sample provided: example as hospital admission or workday).
Indicate the items that are part of each of the factors in the questionnaire with 11 items.
Author Response
Dear Editor in Chief and reviewers of the International Journal of Environmental Research and Public Health. Thank you for your review of manuscript entitled “The Nursing Stress Scale - Spanish Version: an update to its psychometric properties and validation of a short-form version in acute care hospital settings”.
Firstly, I would like to thank you for considering the manuscript for peer review. Similarly, I would like to thank you and the reviewers for their valuable contributions and comments made regarding the above-mentioned manuscript. All suggestions have been carefully considered and all modifications have been made as necessary.
Now, we answer your suggestions in detail:
Review 1
The manuscript presents the results of a study aimed at updating the psychometric properties of the “Nursing Stress Scale - Spanish version” and validating a short-form version in nursing staff in acute care hospital settings in the Public Health Service in Spain. The manuscript is well structured.
The topic is relevant since the valid tools can be extremely useful in improving quality of worker especially for those who work directly with patients. In this sense, it is essential to have tools that allow us to quickly evaluate those constructs that we want to measure.
However, in my opinion, the study presents some methodological problems that need to be addressed and require reorientation on the part of the researchers. I detail these problems below.
Materials and Methods
You have indicated how recruited the participants for approximately two years, guaranteeing the anonymity of the participants. It does not indicate how it has guaranteed that in that period of time the same nurse or graduate has not participated in the study twice.
Authors: Thank you for the suggestion. We have included more information about this topic in Materials and Methods section (line 150-152).
Line 161 describes that the NSS instrument is composed of seven areas, but does not indicate the number of items in each area.
Authors: Thanks to your advice. We have included all information about the number of items in each area (line 169).
I think it is good that they have carried out an exploratory factor analysis of the 34-item questionnaire but it would be more accurate to include a confirmatory factor analysis. The structure of this questionnaire is well known and it is important to see how it behaves in the study population in order to answer the first objective.
Authors: We fully agree that was necessary included. Then, we have carried out a Confirmatory Factor Analysis and included this information in Materials and Methods section.
Results
Specify in table 1, if there are missing data (the sum of some of the variables does not give the total sample provided: example as hospital admission or workday).
Authors: Thank you for your advice. We have detected some mistakes and we have modified the data in the table 1.
Indicate the items that are part of each of the factors in the questionnaire with 11 items.
Authors: This information have been indicated in table 3.
Thank you for your time and consideration.
Sincerely, the authors.
Reviewer 2 Report
This is an article meeting very high technical standards. It is of valuable applicability, provides a very convincing and concise introduction,
and a very sound classical test theory analysis, including reliability of subscales (factors) and the validation of an 11-items short form.
The sampling with its multi-center approach is particularly appreciable, as well as its documentation. The discussion gives an outlook
exceeding the mere technical aspects, in addition to the comparison with results in other countries.
In an additional study, an item response theory-based analysis might add on some information.
Suggestions:
p. 2, 82: please indicate to what criterion "explained variance" refers to
Some non-responder analysis is recommended (the article mentions the response rate 52.5% as a limitation), which should be doable
at least per institution
Author Response
Dear Editor in Chief and reviewers of the International Journal of Environmental Research and Public Health. Thank you for your review of manuscript entitled “The Nursing Stress Scale - Spanish Version: an update to its psychometric properties and validation of a short-form version in acute care hospital settings”.
Firstly, I would like to thank you for considering the manuscript for peer review. Similarly, I would like to thank you and the reviewers for their valuable contributions and comments made regarding the above-mentioned manuscript. All suggestions have been carefully considered and all modifications have been made as necessary.
Now, we answer your suggestions in detail:
Review 2
This is an article meeting very high technical standards. It is of valuable applicability, provides a very convincing and concise introduction, and a very sound classical test theory analysis, including reliability of subscales (factors) and the validation of an 11-items short form.
The sampling with its multi-center approach is particularly appreciable, as well as its documentation. The discussion gives an outlook exceeding the mere technical aspects, in addition to the comparison with results in other countries.
In an additional study, an item response theory-based analysis might add on some information.
Suggestions:
- 2, 82: please indicate to what criterion "explained variance" refers to
Authors: Thank you for the suggestions, we have modified in the manuscript.
Some non-responder analysis is recommended (the article mentions the response rate 52.5% as a limitation), which should be doable at least per institution
Authors: We have been included data about non-respondent profile in the text (line 150-152).
We hope that the modifications meet your expectations and that you can continue to consider the manuscript for publication in your journal.
Thank you for your time and consideration.
Sincerely, the authors.
Reviewer 3 Report
This manuscript is well written and presents the validation of the NSS (Nursing Stress Scale) among a large sample of nurses in Spain. I agree with the authors that measures should be regularly checked in terms of their psychometric properties. The authors present a carefully executed and clearly described sampling procedure, and report on an impressive number of participants. I am quite positive about this manuscript, but also have some remarks which should be addressed:
- The non-response rate is quite high (48.5%); did authors perform any analyses to gain insight into these non-responders? Is anything known about their demographics or reasons for not completing the survey? It is also known from research that responders and non-responders differ quite substantially in several domains (see e.g. Cheung et al. 2017), which could - as the authors also mention on page 11 - have biased the current results. I feel some more information about the non-responders is needed.
- The division between RN and LPN is consistently made throughout the manuscript, but the reasons for doing so are not clear to me. More background for non-nurses is needed, and/or about the Spanish context. Why would these groups differ? I assume that such proposed differences are the reason for the separate analyses?
- The authors mention that they removed the item 'breakdown of computer' (see page 4) and replaced it with another item ('frequent job interruptions'). I wonder why this decision was made, why this particular item was removed (without basing this on psychometric properties), and why the replacement item was chosen? It seems to me that in a validation study, these decisions cannot be made before administering the questionnaire, but should be based on the outcome of the psychometric analyses.
- I was quite surprised that the authors conducted EFA to validate the measure(s); given that the measure had already been validated and the current manuscript provides an 'update', I had expected maximal likelihood Confirmatory Factor Analysis to verify the underlying factor structure. The approach reported by e.g. Montero-Marin et al. (2014) is more rigorous and could be used as an example for the current study.
- Additional indications for convergent validity (e.g. with burn-out symptoms) would have been informative; the lack of this could be addressed in the discussion section.
- The validation of the short-form (11 item) measure is described quite concisely; for example, why only base decisions on factor loadings <.5 and not on the highest corrected item-total correlation within each subscale?
- I think the readability of the discussion would be improved if the COSMIN report would be in a table, and in the results section.
References
Cheung, K.L., ten Klooster, P.M., Smit, C. et al. The impact of non-response bias due to sampling in public health studies: A comparison of voluntary versus mandatory recruitment in a Dutch national survey on adolescent health. BMC Public Health 17, 276 (2017). https://doi.org/10.1186/s12889-017-4189-8
Montero-Marin, J., Piva Demarzo, M. M., Pereira, J. P., Olea, M., & García-Campayo, J. (2014). Reassessment of the psychometric characteristics and factor structure of the ‘Perceived Stress Questionnaire’(PSQ): analysis in a sample of dental students. PloS one, 9(1), e87071.
Author Response
Dear Editor in Chief and reviewers of the International Journal of Environmental Research and Public Health. Thank you for your review of manuscript entitled “The Nursing Stress Scale - Spanish Version: an update to its psychometric properties and validation of a short-form version in acute care hospital settings”.
Firstly, I would like to thank you for considering the manuscript for peer review. Similarly, I would like to thank you and the reviewers for their valuable contributions and comments made regarding the above-mentioned manuscript. All suggestions have been carefully considered and all modifications have been made as necessary.
Now, we answer your suggestions in detail:
Review 3
This manuscript is well written and presents the validation of the NSS (Nursing Stress Scale) among a large sample of nurses in Spain. I agree with the authors that measures should be regularly checked in terms of their psychometric properties. The authors present a carefully executed and clearly described sampling procedure, and report on an impressive number of participants. I am quite positive about this manuscript, but also have some remarks which should be addressed:
The non-response rate is quite high (48.5%); did authors perform any analyses to gain insight into these non-responders? Is anything known about their demographics or reasons for not completing the survey? It is also known from research that responders and non-responders differ quite substantially in several domains (see e.g. Cheung et al. 2017), which could - as the authors also mention on page 11 - have biased the current results. I feel some more information about the non-responders is needed.
Authors: Thanks for all your comments. We have included more information about non-respondent profile in manuscript (line 150-152).
The division between RN and LPN is consistently made throughout the manuscript, but the reasons for doing so are not clear to me. More background for non-nurses is needed, and/or about the Spanish context. Why would these groups differ? I assume that such proposed differences are the reason for the separate analyses?
Authors: Thanks to your suggestion, we have improved the explanation and differences between RN and LPN (line 122-125).
The authors mention that they removed the item 'breakdown of computer' (see page 4) and replaced it with another item ('frequent job interruptions'). I wonder why this decision was made, why this particular item was removed (without basing this on psychometric properties), and why the replacement item was chosen? It seems to me that in a validation study, these decisions cannot be made before administering the questionnaire, but should be based on the outcome of the psychometric analyses.
Authors: Thank you for your advice, this was a mistake. Authors than replaced this items were “Escribà-Agüir, V; Más-Pons, R.; Cardenas, M.; Pérez, S. Validación de la escala de estresores laborales en personal de enfermería: «the nursing stress scale». Gac Sanit. 1999,13, 191–200”. In this paper, authors have used the version validated by this authors.
I was quite surprised that the authors conducted EFA to validate the measure(s); given that the measure had already been validated and the current manuscript provides an 'update', I had expected maximal likelihood Confirmatory Factor Analysis to verify the underlying factor structure. The approach reported by e.g. Montero-Marin et al. (2014) is more rigorous and could be used as an example for the current study.
Authors: We fully agree that was necessary included. Then, we have carried out a Confirmatory Factor Analysis and included this information in Materials and Methods section.
Additional indications for convergent validity (e.g. with burn-out symptoms) would have been informative; the lack of this could be addressed in the discussion section.
Authors: Thanks for your comments. We have extended the information in Discussion section (line 333-334).
The validation of the short-form (11 item) measure is described quite concisely; for example, why only base decisions on factor loadings <.5 and not on the highest corrected item-total correlation within each subscale?
Authors: Thanks to your suggestion, we have improved this information in the manuscript (line 199).
I think the readability of the discussion would be improved if the COSMIN report would be in a table, and in the results section.
Authors: Thanks for your comments. We have included the COSMIN checklist in the results section (line 263).
We hope that the modifications meet your expectations and that you can continue to consider the manuscript for publication in your journal.
Thank you for your time and consideration.
Sincerely, the authors.
Round 2
Reviewer 3 Report
Thank you for revising your manuscript so thoroughly. I feel the scientific value has greatly improved, and the addition to the literature is clearer now. I think the authors presented their work in an understandable way and I recommend publication at this point.